# SAR and Optical Data Applied to Early-Season Mapping of Integrated Crop–Livestock Systems Using Deep and Machine Learning Algorithms

Ana P. S. G. D. D. Toro [1,*] , Inacio T. Bueno [1,2] , João P. S. Werner [1] , João F. G. Antunes [3] ,
Rubens A. C. Lamparelli [1,2] , Alexandre C. Coutinho [3] , Júlio C. D. M. Esquerdo [1,3] , Paulo S. G. Magalhães [2]
and Gleyce K. D. A. Figueiredo [1]

1   School of Agricultural Engineering, University of Campinas, Campinas 13083-875, SP, Brazil
2   Interdisciplinary Center of Energy Planning, University of Campinas, Campinas 13083-896, SP, Brazil
3   Embrapa Digital Agriculture, Brazilian Agricultural Research Corporation, Campinas 13083-886, SP, Brazil
*   Correspondence: anapaolagomes@hotmail.com

**Abstract:** Regenerative agricultural practices are a suitable path to feed the global population. Integrated Crop–livestock systems (ICLSs) are key approaches once the area provides animal and crop production resources. In Brazil, the expectation is to increase the area of ICLS fields by 5 million hectares in the next five years. However, few methods have been tested regarding spatial and temporal scales to map and monitor ICLS fields, and none of these methods use SAR data. Therefore, in this work, we explored the potential of three machine and deep learning algorithms (random forest, long short-term memory, and transformer) to perform early-season (with three-time windows) mapping of ICLS fields. To explore the scalability of the proposed methods, we tested them in two regions with different latitudes, cloud cover rates, field sizes, landscapes, and crop types. Finally, the potential of SAR (Sentinel-1) and optical (Sentinel-2) data was tested. As a result, we found that all proposed algorithms and sensors could correctly map both study sites. For Study Site 1(SS1), we obtained an overall accuracy of 98% using the random forest classifier. For Study Site 2, we obtained an overall accuracy of 99% using the long short-term memory net and the random forest. Further, the early-season experiments were successful for both study sites (with an accuracy higher than 90% for all time windows), and no significant difference in accuracy was found among them. Thus, this study found that it is possible to map ICLSs in the early-season and in different latitudes by using diverse algorithms and sensors.

**Keywords:** regenerative agriculture; transformer; LSTM; random forest; multisource; ICLS

## 1. Introduction

Currently, regenerative agriculture practices are essential for scaling up food production to feed the growing world population while avoiding substantial environmental impacts [1]. Integrated crop–livestock systems (ICLSs) are a common practice of regenerative agriculture with synergy between crops and livestock [2]. Based on a spatial–temporal integration of crop and livestock, ICLSs are a promising alternative to reaching sustainable food production through land-use diversification. Thus, it aligns with the Sustainable Development Goal (SDGs): food security (SDG2—Zero Hunger), the mitigation of environmental impacts (SDG13—Climate Action), and land conservation (SDG15—Life on Land) [3,4].

Due to the growing adoption of ICLSs in Brazil, driven by the Plan for Low Carbon Emission in Agriculture—ABC Plan and ABC Plan+, a systematic monitoring system is needed to identify new areas and monitor the existing sites. However, the main challenge in remotely identifying and monitoring this type of practice is the highly dynamic nature of the system, which results from the succession of different land uses and management [5].

The use of remote sensing data for agricultural monitoring increased significantly with the growing availability of high temporal, spatial, and spectral data [6]. Previous studies succeeded in identifying ICLSs using optical data [7–9] once it presented detailed information about the vegetation canopy. However, some factors decrease its applicability for agricultural monitoring, such as the high cloud coverage in tropical regions and high rates of pixel saturation [10–14]. Thus, synthetic aperture radar (SAR) data is an alternative for tropical regions to use in land-use monitoring due to its all-weather observation capabilities [15–17]. In addition, the target orientation affects the SAR signal, generating additional information about the physical structure based on different signal polarizations [18].

Remote sensing satellite image time series (SITS) are efficient ways to map and monitor areas with ICLS [19], mainly due to the high temporal dynamics of these systems. However, there has been no detailed investigation of the optical and SAR data used to identify ICLSs. Moreover, the studies have only dealt with a time window composed of the entire crop season as input to reduce the high complexity of the time series [20]. These approaches can limit the application of the methodologies for in-season purposes while consuming more time and resources [21]. According to the annual report of the European Commission for agricultural monitoring, the success of agricultural monitoring consists of accurately mapping the early season of crops. [22–24]. Having this information before the end of the season could be essential for the decision-making process [25]. Nonetheless, differentiating crops without an entire season's time series could be challenging, mainly when crops share the same growing season [26].

In this context, deep learning algorithms have been considered a breakthrough approach in the research field of remote sensing. They represent the state-of-the-art for crop-type mapping. Deep architectures, such as recurrent neural networks—RNN and transformers, have recently attracted broad attention for the handling of highly complex information in time series analyses. They tend to perform better with reduced data. The RNNs are capable of dealing with data sequences in such a way that the output of the previous time-step is the input data for the current step [27], allowing the architecture to handle temporal problems [28]. Vaswani et al. [29] proposed the transformer architecture for natural language processing based on a sequential analysis. It can combine multiple self-attention layers with short connections [30]. The self-attention network can handle temporal data and manage complex inputs. However, there is very little published research on testing this network for crop mapping [31–33] although promising results have been achieved for a small number of studies.

In this context, this study explores the potential of Sentinel-1 and Sentinel-2 data, deep and machine learning algorithms, and time windows in different study areas to map ICLSs, addressing the following research questions: (i) is it suitable to perform an ICLS mapping in different site locations with a common methodology? (ii) How do sensor type, algorithm architecture, and time window size affect classification accuracy? Thus, the objectives of this study were to map ICLS fields at two different site locations, using SAR and optical data, deep learning algorithms, and early-season-based detection, and to assess significant differences in the obtained accuracy in each location, sensor, algorithm, and time window.

## 2. Materials and Methods

### 2.1. Study Sites

Two study sites were selected to evaluate the performance of different sensors and algorithms to map the early-season of regenerative agricultural practices. In this study, we focused on the practice of integrating crops and livestock in a dynamic system. The ICLS has three main objectives, (i) reduce the soil cyclical nutrient loss and consequently increase plant productivity, (ii) organize agricultural practices in such a way that contributes to the ecosystem services, and (iii) increase resilience from an economic and environmental point of view [34]. The integrated system could be composed of agriculture, forestry, and livestock activities, and those activities could occur as intercropping, crop rotation, or crop succession [35]. In Brazilian agriculture, the main crops usually present in ICLSs

are soy, corn, and rice, typically followed or consorted by pasture [36]. Thus, the broad range of possibilities regarding the ICLS leads to a very dynamic system with different configurations depending on the approach selected by the farmer. The study sites were dispersed across the Brazilian territory and presented distinct edaphoclimatic conditions and crop types. For both study sites, the one year season period ranged from September to August.

Study Site 1 (SS1) was a farm located in the western portion of Sao Paulo state. It had surrounding fields in the municipality of Caiuá, at coordinates 21°36′26.3″S and 51°51′57.9″W (Figure 1a). The property had an area of 2033.15 hectares, and the soils were predominately sandy loam Ultisols. The region has a climate Aw type, corresponding to tropical climate conditions with a dry season in the winter (June–August) and a rainy season in the summer (December–April) [37]. The average temperature of the region is 24.1 °C, and the average annual precipitation is 1496 mm [37].

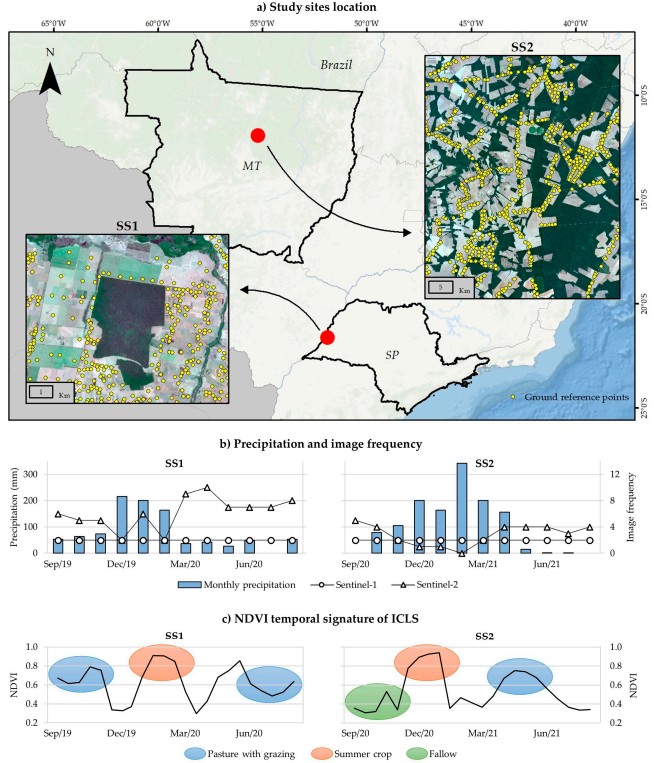

**Figure 1.** (**a**) Location of study sites and ground reference points; (**b**) frequency of Sentinel-1 and Sentinel-2 images used and precipitation rates over the season by study site; (**c**) NDVI temporal signatures of ICLS samples in Mato Grosso and Sao Paulo states, Brazil. The NDVI values were extracted from a sampled pixel inside an ICLS field in each study site. To better illustrate the pattern, the Savitzky–Golay filter was applied.

Study Site 2 (SS2) was located in the north portion of Mato Grosso state and comprised the Santa Carmem and Sinop municipalities at coordinates 11°55′9.40″S and 55° 6′16.52″W (Figure 1a). According to the Koppen classification, the region has an Am type (short dry season) [37]. The average temperature of the region is 30 °C, and the average annual precipitation ranges from 1800 to 2300 mm and is considered the wettest portion of the Aw climate in Brazil [37].

At SS1, the process of implementing the ICLS started in 2013 [2]. According to the farm owners, the main objectives for the implementation of the ICLS were the recovery and improvement of pastures with a cost reduction, the conservation of soil, increasing the amount of soil organic matter, having productive pastures in the winter (dry), reducing the consumption of animal feed, and diversifying the production system and invoicing/income.

In this ICLS, soybean and forage species (*Brachiaria* and *Panicum*) were inter-cropped in the first and second seasons.

In the SS2, some farms had been producing cattle since 1989. Since 2004, some producers adopted ICLS, which resulted in a 3.7% higher productivity than the producers who used a mono-culture approach [38]. For SS2, the ICLS fields were composed of corn, followed by pasture, or soybeans, followed by pasture. Figure 1c exposes the ICLS field behavior in SS1 and SS2. In SS1, the crop season was preceded and succeeded by pasture with grazing, while the crop season in SS2 was preceded by fallow and succeeded by pasture with grazing.

According to the National Supply Company (CONAB), the first season corresponded to the summer season (Oct–Mar), and the second season corresponded to the winter season (Apr–Sep) for both sites [39].

The idea to use two study sites goes beyond exploring the method's flexibility for different study areas. Both locations have different crop types. Furthermore, it is essential to highlight that SS1 is very different from SS2 in terms of precipitation rates (Figure 1b), directly impacting the clear optical image acquisition and the SAR backscatter. Finally, the two study sites have different sizes of fields. The fields of SS1 were generally small (mean size of 1.13 hectares). In contrast, since SS2 mainly comprised agricultural areas (cash crops) instead of pasture, it was less segmented and had larger fields (mean size of 10.91 hectares).

### 2.2. Field Data Collection

Figure 1a illustrates the ground reference data distribution for both areas. For both study sites, a field survey was conducted, the fields were visited, and the ground truth label was annotated and geo-referenced. For SS1, 1205 points were collected for the 2019/2020 season. In SS2, 1118 points were collected for the 2020/2021 season. SS1 contained the classes: ICLS (20.7% of samples), native forest (35.4% of samples), pasture (25% of samples), pasture consortium (7.5% of samples), eucalyptus (6.7% of samples), and wet areas (3.9% of samples). SS2 had four classes: ICLS (4.4% of samples), native forest (68% of samples), pasture (13.4% of samples), and double crop (14.1% of samples). We highlight that, for both study sites, the data are imbalanced.

### 2.3. Remote Sensing Data Collection and Preprocessing

Images from the Sentinel-1 C-band SAR and Sentinel-2 multispectral instrument (MSI) satellites, freely available at the Copernicus Open Access Hub, were used to carry out this study. For each study site, the acquisition period was the same as the ground truth data collection. Thus, the acquisition period was from September to August (2019–2020 for SS1, and 2020–2021 for SS2).

The image preprocessing for Sentinel-1 was carried out by the Sentinel Application Platform (SNAP) software offered by the European Space Agency (ESA). The Sentinel-1 constellation comprises two satellites, Sentinel-1 A and Sentinel-1 B, both of which use the C band (in the 5.495 GHz frequency) with a spatial resolution of 10 m and a temporal resolution of 12 days. For this study, we used images from the wide interferometric (IW) acquisition mode, which provides dual polarization (vertical vertical—VV and vertical horizontal—VH) with a processing Level 1 (ground range detected—GRD). First, the images were calibrated to obtain the backscatter coefficients. The noise reduction Lee filter (5 × 5) was then applied. Finally, a terrain correction was performed to correct geometric distortions. The backscatter coefficients were converted to decibels (Db) to facilitate the analysis of the results.

Regarding the preprocessing of Sentinel-2 products: they were obtained at Level- 1C (the top of atmosphere reflectance). To obtain the surface reflectance, the atmospheric correction was conducted using the Sen2Cor algorithm [40]. The product radiometric resolution is 12 bits, with a swath width of 290 km and a wavelength that ranges from 443 nm to 2190 nm. The spatial resolution varies from 10 m to 60 m. All bands with a spatial resolution coarser than 20 m were resampled to 10 m (nearest neighbor) and cropped to the

area of interest. For SS1, images with cloud cover were removed, since only a few images needed to be discarded. On the other hand, for SS2, the scene classification (SC) product from the Sen2Cor package was used to mask cloud and shadow since less than 10% of images would be available with 0% cloud cover [40].

In sequence, vegetation indices are frequently described as sensitive to cropland differentiation, in addition to the fact that they bring essential information about crop phenology and biomass [41]. Thus, the following vegetation indices were calculated for Sentinel-1 and Sentinel-2 images (Table 1).

The Sentinel-2 cited indexes were chosen considering the literature review, which highlighted their potential for crop mapping [8,42]. Regarding the indexes from Sentinel-1, there was an increasing use of radar vegetation indexes for crop-type mapping [18,43].

**Table 1.** Sentinel-1 and Sentinel-2 indexes description.

| Sensor | Index | Equation | Reference |
|---|---|---|---|
| Sentinel-2 | Normalized Difference Vegetation Index—NDVI | $(NIR - RED)/(NIR + RED)$ | [44] |
| Sentinel-2 | Normalized Difference Red Edge Index—NDRE | $(NIR - REDEDGE)/(NIR + REDEDGE)$ $(2.5 \times NIR - RED)/(NIR + 6RED - 7.5BLUE) + 1)$ | [45] |
| Sentinel-2 | Enhanced Vegetation Index—EVI | $(REDEDGE/RED)$ $(VH \times 4)/(VH + VV)$ | [46] |
| Sentinel-2 | RED EDGE 1 | $VH/VV$ | [47] |
| Sentinel-1 | Radar Vegetation Index—RVI | | [48] |
| Sentinel-1 | VH and VV ratio | | [49] |

In addition to the indexes, from Sentinel-2, bands 2 (blue), 3 (green), 4 (red), 6 (red edge 2), 8 (NIR), and 10 (SWIR) were used as inputs. For Sentinel-1, the VV and VH polarizations were used as inputs (We used Band 6 of the Sentinel 2 data as REDEDGE).

### 2.4. Multitemporal Segmentation

The crop-type mapping was conducted at the object level. Thus, segments were generated for both study sites segments. The segments were generated based on the simple non-iterative clustering (SNIC) algorithm [50], which is freely available on the Google Earth Engine Platform [51]. The SNIC is a variation of the super-pixel algorithm. It efficiently groups pixels with similar spectral values and recognizes individual objects [50,52]. For each study site, the temporal interval used to run the algorithm was the same as the one used to acquire ground truth and image data. Both study sites had a dynamic, spatial crop distribution that varied through the season. To address this problem, the temporal median values of NDVI were used. We ran the SNIC algorithm using the following parameters: connectivity: 8; neighborhood size: 80; size: 3; and scale: 5. According to [53], this approach is feasible for obtaining a multitemporal segmentation. Further, for all the available images, the median and standard deviation values for each band and index described in Table 1 were calculated for each polygon and used together as input for the machine and deep learning algorithms.

### 2.5. Machine and Deep Learning Algorithms

We evaluated three algorithms: the random forest (RF), long short-term memory (LSTM) neural network, and transformer (TF) network. The last two are considered deep learning methods, while RF is a machine learning algorithm.

The RF is an ensemble learning algorithm based on decision trees composed of (i) nodes (attributes); (ii) branches (possible attribute values); and (iii) leaves, which are respon sible for identifying the labels for a classification data set [54]. The approach divides the input data into different sets and generates a decision tree for each. Among the advantages of RF, its fast processing speed and stability could be cited, in addition to its easy implementation and robustness [55,56]. This classifier is considered well-established and is broadly applied for crop identification based on satellite data [57,58].

The LSTM neural network was chosen because of its capability to identify the particular temporal behavior of an ICLS [8]. The LSTM network was developed by [59] and is continuously developing. The most straightforward architecture proposed by the authors contains one input, one hidden, and one output layer. The hidden layer contains memory cells and corresponding gate units [60,61]. Compared with custom RNNs, the main idea behind LSTMs is that an adapted cell tends to detect long-term dependencies between variables, in addition the fact that it converges faster than the usual RNNs [62]. The LSTM cell comprises one main layer and the other three gate controllers.

In LSTM cells, there is a main layer that has the principal purpose of analyzing the inputs (x) and the previous state (in the short-term), h(−1). The other three layers work as gates and have binary outputs for opening and closing. Those values are generated through logistic regression [62]. In this context, there is the so-called "forget gate" that controls which parts of the long-term state should be forgotten [63]. Next is the input gate, which controls which parts of the current input should be added to the long-term memory [63]. Finally, there is the output gate, which is responsible for usefully extracting information from the current cell state for being presented as an output [63]. Thereby, we believe that the LSTM's particular approach tends to understand the correlation between the occurrence of different crops in the same field (such as in the case of an ICLS), differentiating this from pasture or double crop, for example. Additionally, considering the early-season crop-type mapping experiment, which will be conducted in this study, LSTM already demonstrated an ability to identify crops before the end of the season [64].

The transformer architecture is considered state-of-the-art for some applications [65]. It has presented a high accuracy for crop-type mapping [66]. It comprises a series of encoder-decoder structures, in which each encoder has self-attention, and normalization layers, which are followed by a feed-forward net [29]. The self-attention layers aim to identify the regions with the most relevant features and thus indicate that the model needs to pay attention to them [29]. Furthermore, this model automatically inputs positional encoders for each input. The concept of multi-head self-attention was implemented, allowing the model to pay attention to different regions at the same time [67]. Garnot et al. [31] used a multi-head, self-attention approach to identify winter and summer cereal seasons. This approach was helpful for paying attention to different parts of the crop cycle. Thus, we believe that the multi-head attention mechanism may be able to identify the difference between sequential annual crops and annual crops consorted with the pastures [30]. Still, the transformer may adapt its attention mechanism differently for each time window, making it feasible to work well with the early-season approach.

For all different approaches, the dataset was partitioned into 70% for training, 10% for validation, and 20% for testing. To avoid bias in the classification, we did not use polygons from the same field in more than one sampling category. Furthermore, the test dataset had never been seen before by the algorithm. To evaluate the potential of early-season classification for ICLS at both study sites, we tested three time windows: (i) the entire season (September to August), (ii) nine months of data (September to June), and (iii) seven months of data (September to April).

For the RF, the default value of 1000 n tress was used. The architectures used for the LSTM and Transformer models are illustrated in Figure 2.

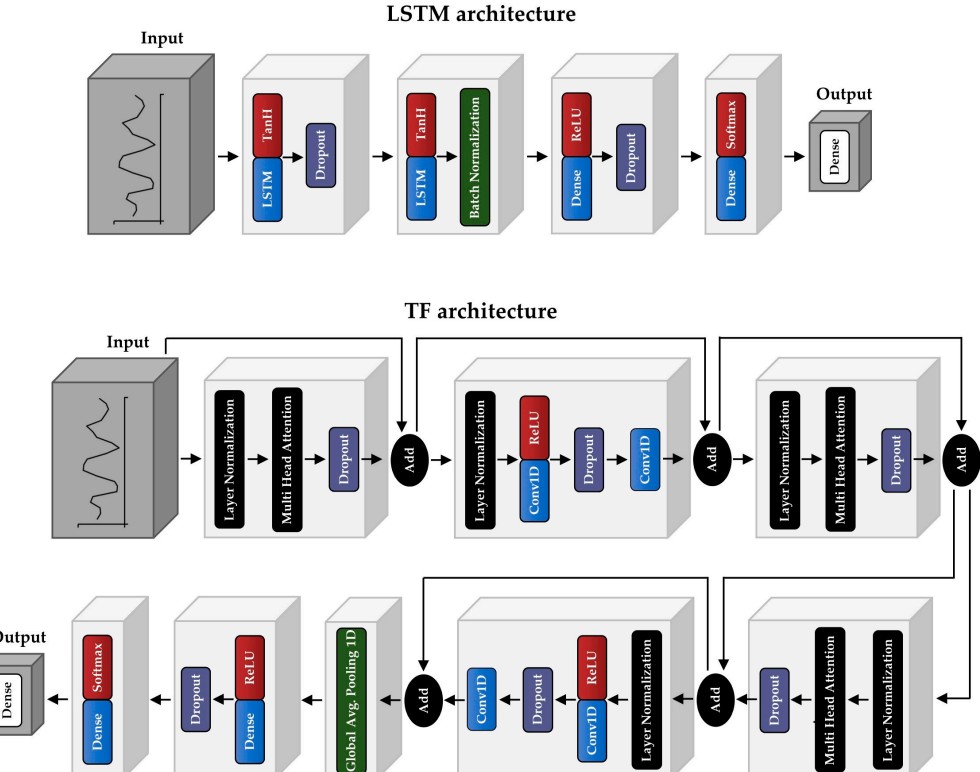

**Figure 2.** LSTM and TF architectures used in models.

Finally, to deal with the imbalanced dataset, synthetic samples were generated using the synthetic minority oversampling technique (SMOTE) algorithm [68], with the kneighbors parameter equal to eight.

### 2.6. Results Evaluation

For both study sites, we analyzed three groups of accuracy based on our method steps: (i) sensor accuracy, (ii) algorithm accuracy, and (iii) time-window accuracy. To identify the statistical significance of the difference among the groups, the Fisher test (F-test) was used and the p-values were generated [69,70]. Thus, if the difference in the accuracy average was at the 5%, level, it was considered significant. For the experiments conducted for this study, the critical values of the F-distribution were F(1,35):4.121, F(1,16):4.49, and F(2,15):3.68 [71]. Therefore, when the F value was higher than its correspondent critical value F-value it was considered that the statistical variation was meaningful. Further, some model predictions were qualitatively evaluated by visual inspection.

## 3. Results

The results showed that there was a significant difference between accuracy in SS1 and SS2 (Figure 3a). The F value was the highest (39.3) when compared with all other F values (Figure 3). This reinforces our hypothesis that both study sites were different even with inputs from the same source, using the same time windows and architectures.

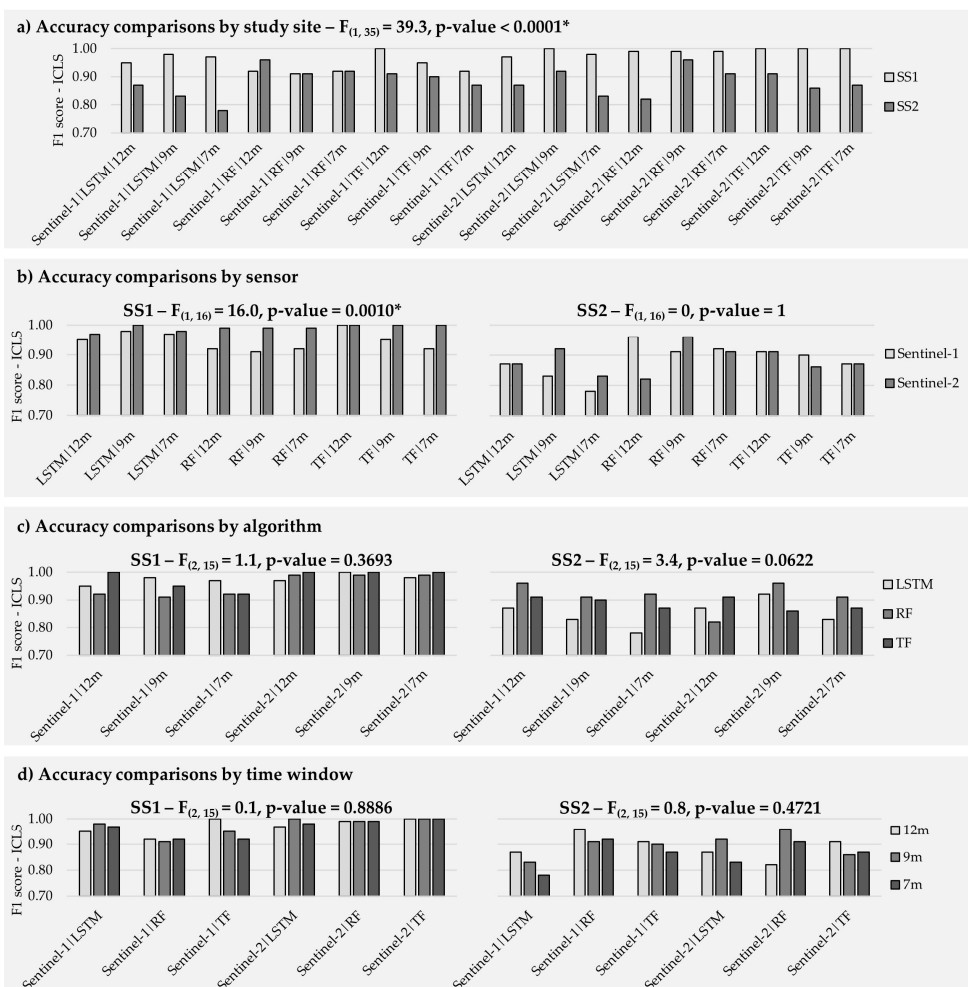

**Figure 3.** Comparisons of F1-score for ICLS, accordingly to study site, sensor, algorithm, and time window. (*) Indicates statistical significance.

### 3.1. Sensor Accuracy

Regarding the performance of both sensors, the overall accuracy of the optical data for SS1 was 9% higher than using the SAR data as an input (Table 2). In contrast, for SS2, the optical data produced an accuracy 1% higher than the SAR data (Table 2). For the ICLS, the results based on Sentinel-2 data were 5% higher than the SAR data in SS1 and were the same for SS2 (Table 2). This indicates that the ICLS accuracy difference between sensors is relevant for SS1 (F value > 3.68) but not for SS2 (F value < 3.68) (Figure 3a,b).

The higher accuracy was achieved by Sentinel-2 for SS1, while the overall accuracy for SS2 using Sentinel-1 was slightly better. However, the F1-scores for either sensor did not show a clear difference. (Table 2).

For the early-season analysis, the time reduction did not affect the overall accuracy of the Sentinel-1 and Sentinel-2 results (Table 2). For the target class, the mean Sentinel 1 accuracy increased with the size of the time window (0.93 for Sep–Aug; 0.91 for Sep–Jun; and 0.89 for Sep–Apr). For Sentinel-2 products, the target class accuracy was not affected, with the mean accuracy being 0.92 for the Sep–Aug period, 0.95 for Sep–Jun, and 0.93 for Sep–Apr (Table 2).

**Table 2.** Overall and ICLS-focused accuracy values for each study area, sensor, classifier, and time interval.

| Study Site | Algorithm | Sensor | Time Window | F1 Score—Overall | Precision—ICLS | Recall—ICLS | F1 Score—ICLS |
|---|---|---|---|---|---|---|---|
| | RF | Sentinel—1 | Entire Season | 0.87 | 0.92 | 0.92 | 0.92 |
| | RF | Sentinel—1 | Sep–Jun | 0.86 | 0.94 | 0.87 | 0.91 |
| | RF | Sentinel—1 | Sep–Apr | 0.87 | 0.9 | 0.95 | 0.92 |
| | RF | Sentinel—2 | Entire Season | 0.98 | 1.00 | 0.98 | 0.99 |
| | RF | Sentinel—2 | Sep–Jun | 0.98 | 1.00 | 0.98 | 0.99 |
| | RF | Sentinel—2 | Sep–Apr | 0.97 | 1.00 | 0.98 | 0.99 |
| | LSTM | Sentinel—1 | Entire Season | 0.89 | 0.97 | 0.94 | 0.95 |
| | LSTM | Sentinel—1 | Sep–Jun | 0.88 | 1.00 | 0.97 | 0.98 |
| | LSTM | Sentinel—1 | Sep–Apr | 0.89 | 0.97 | 0.97 | 0.97 |
| SS1 | LSTM | Sentinel—2 | Entire Season | 0.96 | 0.94 | 1.00 | 0.97 |
| | LSTM | Sentinel—2 | Sep–Jun | 0.97 | 1.00 | 1.00 | 1.00 |
| | LSTM | Sentinel—2 | Sep–Apr | 0.95 | 0.97 | 1.00 | 0.98 |
| | TF | Sentinel—1 | Entire Season | 0.86 | 1.00 | 1.00 | 1.00 |
| | TF | Sentinel—1 | Sep–Jun | 0.85 | 0.97 | 0.94 | 0.95 |
| | TF | Sentinel—1 | Sep–Apr | 0.86 | 0.96 | 0.87 | 0.92 |
| | TF | Sentinel—2 | Entire Season | 0.96 | 1.00 | 1.00 | 1.00 |
| | TF | Sentinel—2 | Sep–Jun | 0.97 | 1.00 | 1.00 | 1.00 |
| | TF | Sentinel—2 | Sep–Apr | 0.95 | 1.00 | 1.00 | 1.00 |
| | RF | Sentinel—1 | Entire Season | 1.00 | 1.00 | 0.92 | 0.96 |
| | RF | Sentinel—1 | Sep–Jun | 0.99 | 1.00 | 0.83 | 0.91 |
| | RF | Sentinel—1 | Sep–Apr | 0.99 | 0.92 | 0.92 | 0.92 |
| | RF | Sentinel—2 | Entire Season | 0.98 | 0.9 | 0.75 | 0.82 |
| | RF | Sentinel—2 | Sep–Jun | 0.99 | 0.92 | 1.00 | 0.96 |
| | RF | Sentinel—2 | Sep–Apr | 0.99 | 1.00 | 0.83 | 0.91 |
| | LSTM | Sentinel—1 | Entire Season | 0.99 | 0.91 | 0.83 | 0.87 |
| | LSTM | Sentinel—1 | Sep–Jun | 0.98 | 0.83 | 0.83 | 0.83 |
| | LSTM | Sentinel—1 | Sep–Apr | 0.98 | 0.82 | 0.75 | 0.78 |
| SS2 | LSTM | Sentinel—2 | Entire Season | 0.98 | 0.91 | 0.83 | 0.87 |
| | LSTM | Sentinel—2 | Sep–Jun | 0.99 | 0.92 | 0.92 | 0.92 |
| | LSTM | Sentinel—2 | Sep–Apr | 0.97 | 0.83 | 0.83 | 0.83 |
| | TF | Sentinel—1 | Entire Season | 0.97 | 1.00 | 0.83 | 0.91 |
| | TF | Sentinel—1 | Sep–Jun | 0.98 | 1.00 | 0.83 | 0.90 |
| | TF | Sentinel—1 | Sep–Apr | 0.97 | 0.91 | 0.83 | 0.87 |
| | TF | Sentinel—2 | Entire Season | 0.97 | 1.00 | 0.83 | 0.91 |
| | TF | Sentinel—2 | Sep–Jun | 0.97 | 1.00 | 0.75 | 0.86 |
| | TF | Sentinel—2 | Sep–Apr | 0.96 | 0.91 | 0.83 | 0.87 |

*3.2. Algorithm Accuracy*

The architecture of the three algorithms did not vary (F < 3.68) considerably between different study sites, time windows, or sensors (Figure 3c).

All algorithms achieved a high accuracy (higher than 0.85) for both study sites and all time windows (Table 2). Regarding the overall accuracy, the RF and LSTM algorithms reached the highest mean value (0.95), slightly higher than the TF model (0.93). However, for our target class, TF and RF algorithms achieved a higher accuracy value (0.93), followed by LSTM (0.91). The three architectures had the same lowest accuracy value (0.75) for ICLS recall metric in SS2, at different time intervals and using different sensors.

None of the algorithms had a significant impact on accuracy due to the time window reduction. The overall accuracy varied by approximately 1% for all algorithms (Table 2). For the ICLS class RF, the TF algorithm maintained the same variance from the entire season to Sep–Apr. However, TF had a higher difference (−4%) (Table 2).

### 3.3. Early Season

For the overall accuracy and ICLS f1-score, the mean accuracy did not change for the Sep–Aug and Sep–Jun time windows (Table 2). For the Sep–Apr time window, the mean overall accuracy was 0.01 lower. For the target class, it was, on average, 0.03 lower (Table 2). Figure 3d demonstrates that there was no statistical difference among the time windows for any of the study sites. However, when using LSTM for SS2, Sentinel-1 achieved lower accuracies, even though the entire season achieved an F1-score of 0.87 (Figure 3d). Moreover, SS1 was less impacted by the time window reduction than SS2, in which all results outperformed the 0.90 F1 scores for the ICLS (Figure 3d).

### 3.4. Predictions

Considering the most suitable algorithm (the algorithm with a higher F1 score and shorter time window) for ICLS for each study site, we ran a prediction to evaluate the practical application of the algorithms (Figure 4). Except for the eucalyptus and wet areas classes in SS1, all values were higher than 0.90. In Figure 4, the confusion matrices are also illustrated. For SS1, the highest confusion occurred between eucalyptus, pasture, and wet areas. The confusion with pasture could be explained because the eucalyptus areas used in training were generally young regrowth areas with a shrub aspect that is similar to certain pasture fields. Further, the confusion of eucalyptus and pasture with wet areas could be due to some training samples having been collected in regions close to the wet regions. Finally, in SS1, the pasture class was barely confused with the pasture consortium. This was expected since both classes contain pasture species.

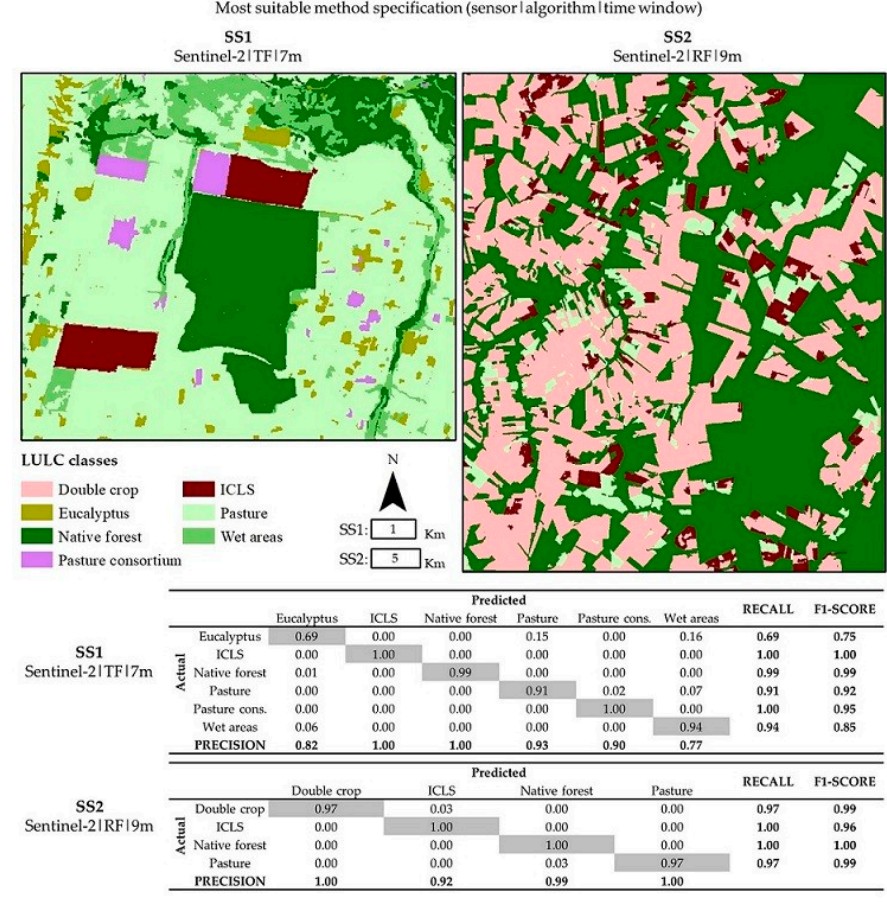

**Figure 4.** More suitable method specifications for ICLS (sensor, algorithm, and time window) for each study site, confusion matrices, and the respective values of precision, recall, and the F1 score for all classes.

For SS2, some double crop fields were classified as ICLS; the algorithm could likely not distinguish between the second crop and the pasture class. In sequence, some pasture fields were classified as native forests. This could be due to an edge effect as native forest areas surrounded most of the pasture fields in SS2.

In addition to the qualitative evaluation of the best models, we ran some predictions using different models to demonstrate the relevance of this type of evaluation in addition to its accuracy values (Figure 5). Firstly, for SS1, using models with similar F1 score values, the LSTM model with a time window of Sep–Apr produced a higher F1-score (0.98) for the ICLS than the LSTM model using the entire season (0.95). However, the ICLS fields were less geometrically defined (Figure 5a,b). For the pasture class, the same situation happened with the RF model: a higher F1 score (0.93) produced a noisier prediction than a lower one (0.90) (Figure 5c,d). In sequence, despite a high F1 score (0.95) for the pasture consortium, which was obtained using Sentinel-1 and RF, the pasture consortium field was not identified. However, while using Sentinel-2, which had a slightly higher F1 score (+0.05), the field was entirely identified (Figure 5e,f).

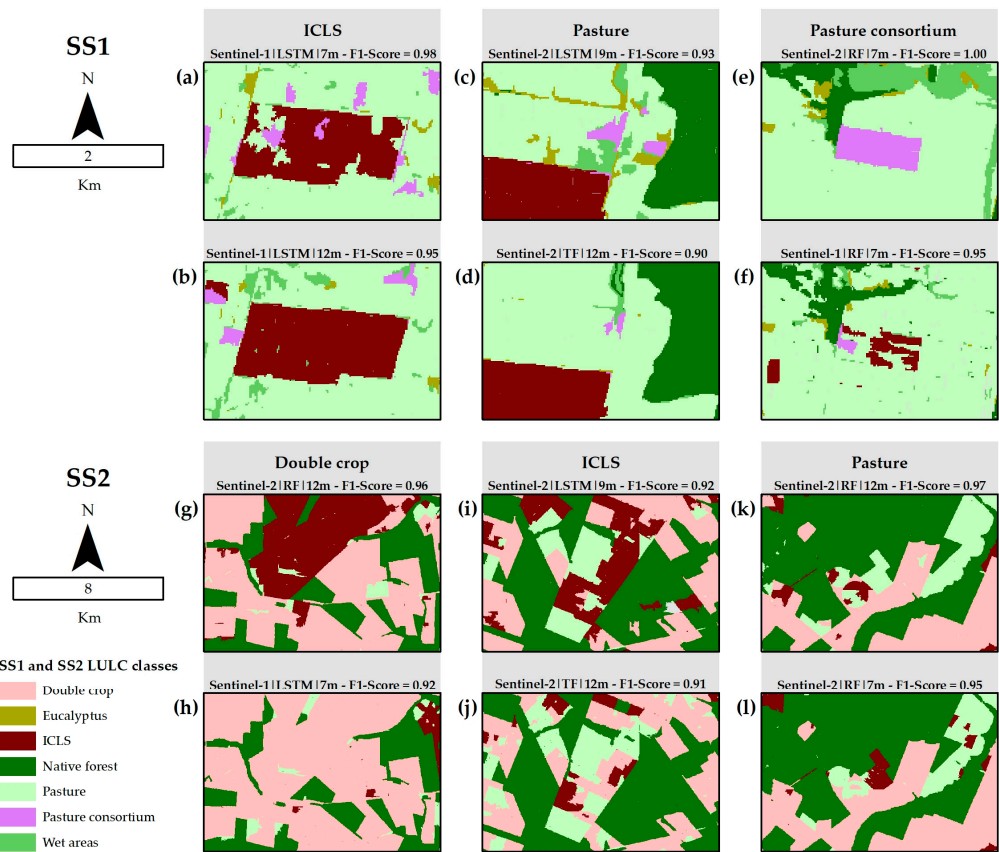

**Figure 5.** Predictions for both study areas using different models and sensors with similar accuracy values.

For SS2, the RF model achieved an accuracy of 0.96 for the double crop class and incorrectly classified it as an ICLS in some regions, while the LSTM achieved a lower accuracy (0.92) without showing such confusion frequently (Figure 5e,f). For the ICLS, the models with similar accuracy could present completely different predictions, as can be observed in Figure 5i,j. Similar to the pasture class, both models, with different time windows, presented a close accuracy (0.97 and 0.95) and very different predictions in certain places (Figure 5k,l).

## 4. Discussion

Despite the differences among the study sites, identifying the ICLS areas using the same methodology was accomplished. This result is promising since many authors reported on the complexity of using use the same model to map different latitudes, which usually implicates different growing periods and crop phenological behavior [72,73]. Further, it should be considered that both areas are surrounded by different landscapes. While SS1 was surrounded by pasture, SS2 was surrounded by double-crop fields and some pasture. Thereby, achieving a high accuracy in both is meaningful since most of the landscape surrounding the ICLS fields in Brazil is composed of pasture or double crop. In addition, it was interesting to see that, even in SS2, where there was a high incidence of clouds, especially during the growing season, it was still possible to identify the ICLS areas using only optical data (RF/Sentinel-2: 0.96).

This study demonstrates that optical data achieved superior results to SAR data for crop-type mapping in agreement with the literature [74]. This could suggest that the rich spectral dimension of Sentinel-2 overcame the temporal resolution due to cloud cover. Thereby, even with some gaps in the time series, the Sentinel-2 classification reached better results. It can be highlighted that, for SS1, it is feasible to work with optical products, especially Sentinel-2, for which there was a tile overlap in the study area. For SS2, SAR products had a higher temporal availability during the primary growing season (November to March) (Figure 1b). However, even with a lack of Sentinel-2 images, it achieved a higher overall accuracy (0.98 for all time windows) than Sentinel-1 (0.91 (Sep–Apr), 0.91 (Sep–Jun), and 0.95 (Sep–Aug)).

The highest variation in accuracy occurred for Sentinel-1 ($-4\%$). This could mean that the spectral information contained in optical data was even more important than the temporal resolution of Sentinel-1. In this context, many authors explored the potential of Sentinel-2 spectral information for mapping crops [75,76]. On the other hand, Sentinel-1 data also generated high-accuracy results and predictions [77]. This indicates that it would be possible to use this data source in case there are no optical data available. Many studies have shown the potential of SAR as input for crop-type mapping [78,79]. Recently, another subject of study has been the synergic use of both sensors for crop-type mapping. This type of approach usually overcomes the use of a single sensor. Thus, it would be interesting to see this analysis conducted using both datasets fused [80,81].

Regarding model performance, the random forest model obtained the same and sometimes higher accuracy than the deep learning models tested, as was identified previously by many studies [82,83]. Thus, as demonstrated, ICLS areas can be identified without complex algorithms. However, for future approaches, the transferability of those models should be tested since some authors indicated that the random forest model tends to perform transfer learning worse than deep learning models [84]. Therefore, inter-year and inter-region model transfer should be conducted to verify the generalization potential of these models when they are aiming at mapping ICLS fields.

Evaluating the early-season results, it is clear that this approach could be used with no meaningful statistical difference in accuracy. This is a relevant finding since early-season maps could support the producers and decision-makers earlier than the entire-season approaches [20]. Fritz et al. (2019) mentioned that the early-season mapping of crops is a current gap in the use of remote sensing applied to crop mapping [85]. Additionally, it was interesting to see that even in SS2, the accuracy remained high, even when there were few images available (Figures 1b and 2). Thus, those results demonstrated the feasibility of mapping an ICLS, even at the beginning of the livestock implementation cycle (i.e., the Sep–Apr time window).

Despite some high accuracies, there is still space for refinement since some classes were erroneously predicted. Noisy predictions occurred for all models, sensors, and time windows, indicating no pattern (Figure 5). On the other hand, since some models tested were early-season ones, in a practical application it would be possible to validate the

predictions in the field if needed. Finally, the importance of evaluating the results beyond the accuracy values, which could hide the prediction uncertainties, was demonstrated.

## 5. Conclusions

This study demonstrated that it is feasible to map ICLSs in regions with different latitudes, cloud cover rates, and diverse land uses surrounding them using remote sensing data and a common methodology. Further, both sensors (optical and SAR) could also produce ICLS maps with a similar accuracy for the overall classification and target class. Regarding the algorithms, it was demonstrated that all machine and deep learning architectures were able to similarly map ICLS fields. Furthermore, using the entire time series was not necessary to achieve high accuracy (higher than 85%): it was demonstrated that the time window size does not affect meaningfully the accuracy.

Additionally, it was essential to expose not only the accuracy values but also the predictions. This supports the idea of estimating the practical implementation of the tested algorithms. This approach demonstrated that even a high accuracy could produce noisy predictions.

Finally, since the early-season analysis was successful, as a next step we suggest that an even shorter interval could be tested to find the earliest date at which it is possible to identify ICLS fields.

**Author Contributions:** Conceptualization, A.P.S.G.D.D.T. and G.K.D.A.F.; methodology, A.P.S.G.D.D.T.; software, A.P.S.G.D.D.T.; field data acquisition, J.P.S.W., J.F.G.A., J.C.D.M.E., A.C.C., R.A.C.L. and P.S.G.M.; formal analysis, A.P.S.G.D.D.T., G.K.D.A.F. and I.T.B.; data curation, A.P.S.G.D.D.T. and J.P.S.W.; writing—original draft preparation, A.P.S.G.D.D.T., G.K.D.A.F. and I.T.B.; writing—review and editing, A.P.S.G.D.D.T., G.K.D.A.F., I.T.B., J.P.S.W., J.F.G.A., J.C.D.M.E., A.C.C., R.A.C.L. and P.S.G.M.; supervision, G.K.D.A.F.; funding acquisition, G.K.D.A.F., J.F.G.A., J.C.D.M.E., A.C.C., R.A.C.L. and P.S.G.M. All authors have read and agreed to the published version of the manuscript.

**Funding:** This research was funded by the Sao Paulo Research Foundation—FAPESP (Grants: 2021/15001-9 and 2017/50205-9).

**Acknowledgments:** The authors are thankful for the National Council for Scientific and Technological Development (CNPQ) by the Research Grant n. 305271/2020-2, and for the Coordination for the Improvement of Higher Education Personnel (CAPES) (Finance Code 001) for the fellowship granted to J. P. S. Werner.

**Conflicts of Interest:** The authors declare no conflict of interest.

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
