# Peer review of "SAR and Optical Data Applied to Early-Season Mapping of Integrated Crop–Livestock Systems Using Deep and Machine Learning Algorithms"

_remotesensing, doi:10.3390/rs15041130_

Round 1

Reviewer 1 Report

The manuscript is well-organized and articulated. It presents a feasible study on the possibilities of mapping ICLS with different techniques and different sensor data. Related issues are discussed and analyzed.

Author Response

Dear Reviewers and Editor, 

We are thankful for the reviewer's suggestions which will undoubtedly contribute to the improvement of the manuscript. All reviewer comments and suggestions were accepted and implemented in the manuscript, highlighted in yellow and detailed below.

The minor reviewers' suggestions were accepted, highlighted in yellow in the main manuscript.

Reviewer 1

We appreciate very much your comments about the manuscript.

Reviewer 2 Report

Please see attached PDF.

Author Response

Dear Reviewers and Editor, 

We are thankful for the reviewer's suggestions which will undoubtedly contribute to the improvement of the manuscript. All reviewer comments and suggestions were accepted and implemented in the manuscript, highlighted in yellow and detailed below.

The minor reviewers' suggestions were accepted, highlighted in yellow in the main manuscript.

Reviewer 2

Language corrections

Dear Reviewer, thank you, all the suggested language corrections were made and are highlighted in yellow.

There should be more description of how the NDVI timeseries were computed.

Thank you for your suggestion, we added a better description of how the plotted NDVI time series was calculated in the caption of Figure 1. As described bellow:

The NDVI values were extracted from a sampled pixel inside an ICLS field in each study site. To better illustrate the pattern, the Savitzky-Golay filter was applied.

Describe what months comprise first and second seasons.

Thank you, we agree with this suggestion, since it is necessary for a better overview of the crop seasons in Brazil. Thus, we added the following text with a reference:

[Lines 115 – 117] For both study sites, accordingly to the National Supply Company (CONAB), the first season corresponds to the summer season (Oct-Mar), and the second season corresponds to the winter season (Apr-Sep) [CONAB-calendario agrícola windowDescription. https://www.conab.gov.br/institucional/publicacoes/outras-publicacoes/item/7694-calendario-agricola-plantio-e-colheita. Accessed:2022-01-30.]

Make note that it’s higher cloud cover impacting clear optical image acquisition. Additionally, the wetter on average conditions will impact SAR backscatter as well. Please make note of this.

Thank you for your comment, this suggestion was accepted and added as:

[Lines 119 – 121] Furthermore, it is essential to highlight that SS1 is very different from SS2 in terms of precipitation rates (Figure 1 - b), directly impacting clear optical image acquisition, and also the SAR backscatter.

The window size of the Lee filter needs to be described

Absolutely, we are sorry to miss this detail during the writing process, it was added in the text:

[Line 148] First, the images were calibrated to 147 obtain the backscatter coefficients, then the noise reduction Lee filter (5x5) was applied.

Converting fractional backscatter coefficients (sigma, gamma) to decibels is not at all unique to Irshad 2019. In fact, it’s such a standard practice it doesn’t require a citation. If citations are needed, there is a plethora of much earlier SAR studies from the 1980s and 1990s that implemented the practice.

Thank you for flagging this, we removed the Irshad citation.

Please review the wording and accuracy of these sentences.

We are sorry for the wording, we rewrote the sentence to make it clear and accurate, as follows:

[Lines 175 – 177] The SNIC is a variation of the Superpixel algorithm. It efficiently groups pixels with similar spectral values and recognizes individual objects. 

Was the SNIC algorithm run on Sentinel-2 temporal median NDVI imagery only? Other Sentinel-2 and Sentinel-1 imagery? The end of this paragraph is very unclear.

Dear reviwer, we identified the gap in this paragraph, we rewrote it and now we expect it is clear:

[Lines 182 – 185] According to [53], this approach is feasible to obtain a multitemporal segmentation. Further, for all the available images the median and standard deviation values for each band and index described in Table 1 were calculated for each polygon and used together as input for Machine and Deep Learning algorithms. 

Explicitly state how suitability was determined.

Indeed, we added the following lines to provide an explicit explanation:

[Lines 296 – 297] Considering the most suitable algorithm (the one with a higher F1-score, and shorter time window) for ICLS for each Study site, we ran a prediction to evaluate the algorithms practical application (Figure 4).

Map agricultural cover classes at different latitudes? Please clarify.

Sure, in this sentence we aimed to highlight that agricultural targets usually present different behaviors depending on their location. Besides a longer description, we also added a reference that clearly explores it.

[Lines 330 – 332] This result is promising since many authors reported how complex it is to use the same model to map different latitudes, which usually implicates different growing periods and crop phenological behavior [72,73].

“other land uses landcover” is a confusing statement. Please modify

Thank you for your comment, considering your suggestion we modified it to:

[Line 333] Further, it should be considered that both areas are surrounded by different landscapes

There needs to be further description about why optical mapping performance is still exceeding the capabilities of SAR even in locations where clouds are so frequently an issue. The study cited here [70] is from Belgium. Are cloud cover probabilities in Brazil similar? Perhaps they are, but there needs to be further description. In locations that tend to be more cloud free, it is indeed expected that optical Sentinel-2 with 10+ bands would provide more accurate crop classifications than Sentinel-1 with generally only two bands (VV and VH).

We appreciate your comment, to address that we added a reference from a study conducted in Brazil, and also we highlighted that possibly the multispectral resolution of S2 is enough to overcome the lack of clear images:

[Lines 339 - 343]This study demonstrates that optical data achieved superior results than SAR data for crop type mapping in agreement with the literature [74], this could suggest that the rich spectral dimension of Sentinel-2 overcomes the temporal resolution due to cloud cover. Thereby, even with some gaps in the time series, the Sentinel-2 classification reaches better results.

344.This is not true. Classified maps can have very high accuracy but still have areas that are misclassified appearing noisy.

Yes, we agree with you. That's the reason we highlighted in the text that even high-accuracy maps could have uncertainties 

[Lines 379 – 381] Finally, it was demonstrated how important it is to evaluate the results beyond the accuracy values, which could hide the prediction uncertainties. 

Reviewer 3 Report

The research was very well designed, addressing important issues in the area of ​​SITS for mapping production systems with the main focus on ICLS.

Below are some questions and suggestions:

-How the time window for the STIS was chosen. Is the early season in the different regions the same? When analyzing figure 1C, it is possible to observe that the temporal windows (ii) and (iii) do not completely cover the grazed pasture, mainly for area 2. What would be critical for area 2, since before annual agriculture is presented by fallow and the integrated pasture is in succession to annual agriculture. As a suggestion, I find it interesting to better present the decision to choose the series from September and not from October or November.

- In Figure 1: c) Was any type of gap-filing filter used to create the time series graph? Indicate whether the graph refers to 1 pixel or the statistics of all ICL points.

-I missed the citation from https://doi.org/10.3390/rs14071648 Especially on line 34;

- In lines 99 and 100 it is not clear which type of summer crop was implemented, even when the authors themselves indicate that the crops are different. I believe it is important to write a little more about these predominant cultivation systems.

- I found the description of field data collection very brief, how it was carried out and what proportion was used for each class. The agro-productive context of the two regions only becomes clear to the reader from the middle to the end of the text. I suggest a short introduction before;

- The segmentation process was carried out using the entire time series of NDVI values, which in theory would group pixels where there was a similar dynamic of crop succession. I didn't quite understand the reason for applying the temporal median. If this decision was made to simplify the process, wouldn't it be better to choose only one date or a few dates for the segmentation?

- What do the authors believe that the iCL class F-score for the RF algorithm applied to Sentinel-2 was lower when applied to the entire SITS? This fact raises my curiosity, because the 7-month series not include the entire phenological cycle of the pasture. The authors indicate the use of the temporal window as a way of simplifying the SITS, as a way to reduce the dimensionality. But what is demonstrated in area 2 is an increase in accuracy when RF is used with shorter series. How do the authors understand what might have happened?

-As a strategy to reduce dimensionality, was the use of phenological metrics considered?

-I missed other indices such as producer and user accuracy to infer about the different configurations there was a higher commission and omission error for the iCL class. I also missed a larger discussion about which class there was more confusion in addition to double cultivation in area 2.

Author Response

Dear Reviewers and Editor, 

We are thankful for the reviewer's suggestions which will undoubtedly contribute to the improvement of the manuscript. All reviewer comments and suggestions were accepted and implemented in the manuscript, highlighted in yellow and detailed below.

The minor reviewers' suggestions were accepted, highlighted in yellow in the main manuscript

Reviewer 3

How the time window for the STIS was chosen. Is the early season in the different regions the same? When analyzing figure 1C, it is possible to observe that the temporal windows (ii) and (iii) do not completely cover the grazed pasture, mainly for area 2. What would be critical for area 2, since before annual agriculture is presented by fallow and the integrated pasture is in succession to annual agriculture. As a suggestion [...]

Dear Reviewer, thanks for highlighting it. We now described better how we choose the time windows (according to the brazilian crop calendar), as follow::

[Lines 115 – 117]  For both study sites, accordingly to the National Supply Company (CONAB), the first season corresponds to the summer season(Oct-Mar), and the second season corresponds to the winter season(Apr-Sep) [CONAB-calendario agrícola window Description. https://www.conab.gov.br/institucional/publicacoes/outras-publicacoes/item/7694-calendario-agricola-plantio-e-colheita. Accessed:2022-01-30.]

I find it interesting to better present the decision to choose the series from September and not from October or November.

Considering that the season starts in September (See CONAB crop calendar), initiating it from September assures that we can take the planting season in different regions.

 In Figure 1: c) Was any type of gap-filing filter used to create the time series graph? Indicate whether the graph refers to 1 pixel or the statistics of all ICL points.

Absolutely, now this information is more clear in the Figure caption:

Figure 1. (a) Location of Study sites and ground reference points; (b) frequency of Sentinel-1 and Sentinel-2 images used and precipitation rates over the season by study site; (c) NDVI temporal signatures of ICLS samples in Mato Grosso and Sao Paulo states, Brazil. The NDVI values were extracted from a sampled pixel inside an ICLS field in each study site to better illustrate the pattern, the Savitzky-Golay(4x4) filter was applied.

I missed the citation from https://doi.org/10.3390/rs14071648 Especially on line 34;

Thank you, this study brings important context for mapping ICLS fields in Brazil,  we added it on line 35 replaced this https://doi.org/10.1016/j.jag.2020.102150 reference by this https://doi.org/10.3390/rs14071648.

In lines 99 and 100 it is not clear which type of summer crop was implemented, even when the authors themselves indicate that the crops are different. I believe it is important to write a little more about these predominant cultivation systems.

Thank you for your suggestion, this relevant information was missing, we added the following sentence:

[Lines 111 - 112] For SS2 the ICLS fields are composed of corn followed by pasture, or soybean followed by pasture.

I found the description of field data collection very brief, how it was carried out and what proportion was used for each class. The agro-productive context of the two regions only becomes clear to the reader from the middle to the end of the text. I suggest a short introduction before;

Thank you for flagging it, we added more details about the ground truth collection, now the paragraph is more complete:

[Lines 127 - 134] Figure 1 - a illustrates the ground reference data distribution for both areas. For both study sites, a field survey was conducted, the fields were visited and the ground truth label was annotated and geo-referenced. For SS1, 1205 points were collected for the 2019/2020 season, and in SS2, 1118 points were collected for the 2020/2021 season. SS1 contained the classes ICLS (20.7% of samples), Native Forest (35.4% of samples), Pasture (25% of samples), Pasture Consortium(7.5% of samples), Eucalyptus (6.7% of samples), and wet areas (3.9% of samples). SS2 had four classes, ICLS (4.4% of samples), Native Forest(68% of samples), Pasture (13.4% of samples), and Double crop (14.1% of samples). We highlighted that for both Study sites the data is imbalanced.

Also, regarding the lack of information about the agricultural context of both study areas, we added a paragraph about ICLS prior to the description of each study site:

[Lines 77 - 87] Two Study sites were selected to evaluate the performance of different sensors and algorithms to map the early season of regenerative agricultural practices. In this study we focused on the practice of integrating crop and livestock in a dynamic system. The ICLS has three main objectives, (i) reduce the soil cyclical nutrients loss and consequently increase plant productivity, (ii) organize agricultural practices in a way that contributes to ecosystem services, and (iii) increase economic resilience to adverse hypotheses from an economic and environmental point of view [32].The Integrated system could be composed of agriculture, forestry, and livestock activities, those activities could occur as intercropping, crop rotation, or crop succession [33]. In Brazilian agriculture, usually, the main crops present in ICLS are soy, corn, and rice, usually followed or consorted by pasture [34]. Thus, the broad range of possibilities regarding ICLS leads to a very dynamic system, with different configurations depending on the approach selected by the farmer. The study sites are dispersed across the Brazilian territory and present distinct edaphoclimatic conditions and crop types. For both Study sites the one-year season period goes from September to August.

The segmentation process was carried out using the entire time series of NDVI values, which in theory would group pixels where there was a similar dynamic of crop succession. I didn't quite understand the reason for applying the temporal median. If this decision was made to simplify the process, wouldn't it be better to choose only one date or a few dates for the segmentation?

Thank you for your comment, we did a temporal median mainly to consider the temporal dynamic of ICLS and get segments as similar as possible from the reality. We modified our text complementing the description:

[Lines 177 - 180] For each Study site, the temporal interval used to run the algorithm was the same as the one used to acquire ground truth and image data. Both Study sites have a dynamic spatial crop distribution that varies through the season. To address this problem, the temporal median values of NDVI were used.

What do the authors believe that the iCL class F-score for the RF algorithm applied to Sentinel-2 was lower when applied to the entire SITS? This fact raises my curiosity, because the 7-month series not include the entire phenological cycle of the pasture. The authors indicate the use of the temporal window as a way of simplifying the SITS, as a way to reduce the dimensionality. But what is demonstrated in area 2 is an increase in accuracy when RF is used with shorter series. How do the authors understand what might have happened?

That is an interesting question, thank you. In fact, we believe that the RF classifier had a superior capacity to deal with the targets than their complexities.  The capacity of RF to deal with complex targets was already demonstrated by different studies as explored in this revision: https://doi.org/10.1016/j.isprsjprs.2016.01.011

As a strategy to reduce dimensionality, was the use of phenological metrics considered?

Thank you for your consideration, we believe that besides phenological metrics already proved to be useful, the Deep Learning algorithms are able to successfully explore crop temporal characteristics and are more flexible in terms of automation and scalability. However, we can consider a further study comparing the achieved results with ones produced using phenological metrics.

I missed other indices such as producer and user accuracy to infer about the different configurations there was a higher commission and omission error for the iCL class. I also missed a larger discussion about which class there was more confusion in addition to double cultivation in area 2.

Thank you for your suggestion, we modified Figure 4 and now it contains the confusion matrices, precision and recall values (commission and omission errors). Further, we also modified the text to discuss the main confusions found:

[Lines 296 - 310] Considering the most suitable algorithm (the one with a higher F1-score and shorter time window) for ICLS for each Study site, we ran a prediction to evaluate the algorithm's practical application (Figure 4). Except for the Eucalyptus and wet areas class in SS1, all values were higher than 0.90. In Figure 4 the confusion matrices were also illustrated. For SS1 the highest confusion occurred between Eucalyptus, Pasture, and Wet areas. The confusion with Pasture could be explained because the Eucalyptus areas used in training are generally young regrowth areas, with a shrub aspect, being similar to certain pasture fields. Further, the confusion of Eucalyptus and pasture with Wet areas could be some training samples collected in regions close to the Wet regions. Finally, in SS1 pasture class was barely confused with pasture consortium, this being expected since both classes contain pasture species.

For SS2, some Double crop fields were classified as ICLS; probably, the algorithm could not distinguish between the second crop and the pasture class. In sequence, some Pasture fields were classified as Native forests. We suppose this could be some edge effect since Native Forest surrounds most pasture fields in SS2.

Round 2

Reviewer 2 Report

The authors have improved the quality of the manuscript significantly. Another English proofreading is suggested however.

The following lines should be changed in the updated version:

11. “Study sites” should be “study sites”
44. Remove "in"
81. "Adverse hypothesis" is incorrect terminology.
341. Specify this is temporal resolution of Sentinel-1 SAR.
359. Change “Regarding the model’s performance” to “Regarding model performance”
393. Replace “noise” with “noisy”

Author Response

Reviewer 2

Dear reviewer, thank you for your suggestions, the corrections were made and are highlighted in yellow as well as the prior ones.

Reviewer 3 Report

I have read the revised version of the manuscript. I still have some conserns about the relationship between the agricultural calendar and the choice of the beginning of the time window (September). In lines 115 and 116, the authors refer to CONAB's agricultural calendar, however, in this calendar the summer crop starts in October, which does not met with the decision to choose SITS to start in September. This divergence should be explained by the authors, considering that the ICLS differ from the other systems mainly by the winter culture.

 I would like to highlight the following points:

(i) In the 12m time window: This decision waives 1 month of winter culture to use 1 month before the summer culture, ie, only 5 months of SITS are used;

(ii) In the 9m time window: 3 months of images are used in the winter culture, which 4 months could be used.

(iii) In the 7m time window: Only 1 month of the winter crop is used, which 2 months could be used.

These questions are fundamental to the manuscript, including to obtain clues on how high accuracy rates were achieved in the detection of a system. The great differential of ICLS is the pasture element present in the winter crop, a period in which it is less privileged in the temporal coverage of the chosen time windows.

I also would like to highlight the best results achieved when only 1 month of winter crops are covered by SITS.

I believe the conclusion that the early season of 7m starting in September, where only 1 month of the winter crop is used, does not affect the classification and in some cases is even better, is incipient. These results raise my doubts about the sample design of the training and validation data, which are little explored in the text.

Also, it is not discussed about the productive units in plots and paddocks, it was approached only point data, where the sample universe is in farms. If we have points used for training and validation in the same paddock, it will be forcing a high accuracy. 

It can also force high accuracy if what precedes ICLS across the property is the same. The way it is presented, makes me believe that there is an importance to the use that precedes the ICL, which is privileged in the spectro-temporal windows to the place of winter culture.

Could it be that in other properties that adopt ICLS, with different interannual rotation systems, these high scores would be achieved?

A data-driven study to detect such complex systems, must densely explore the reality of field cultural practices.

Author Response

Dear Reviewer and Editor, 

We are thankful for the reviewer's suggestions which will undoubtedly contribute to the improvement of the manuscript. All reviewer comments and suggestions were accepted and implemented in the manuscript, highlighted in yellow and detailed in the attached file.
